# The Association Between Job-Seeking & Adult Attention Deficient Hyperactivity Disorder Among Working-Age Individuals in Oman: A Retrospective Study from the National Clinical Service for Adult ADHD in Oman

**DOI:** 10.3390/bs16010067

**Published:** 2026-01-02

**Authors:** Tamadhir Al-Mahrouqi, Omaira Al-Balushi, Salim Al-Huseini, Marwan Al-Battashi, Amira Al-Hosni, Sathiya Murthi Panchatcharam, Samir Al-Adawi, Hassan Mirza

**Affiliations:** 1Department of Behavioral Medicine, College of Medicine and Health Sciences, Sultan Qaboos University, Sultan Qaboos University Hospital, Muscat 123, Oman; 2Psychiatry Residency Training Program, Oman Medical Specialty Board, Muscat 132, Oman; 3Al Masarra Hospital, Ministry of Health, Muscat 111, Oman; 4Research Section, Oman Medical Specialty Board, Muscat 132, Oman

**Keywords:** ADHD, adult, occupational status, job search, Oman

## Abstract

Attention deficit hyperactivity disorder (ADHD) presents significant challenges in all age groups, affecting various aspects of daily functioning and quality of life. Objective: This study aims to explore the rate and associated factors of unemployment among ADHD in adults. A retrospective cohort study was conducted, including 179 adults diagnosed with ADHD seeking consultation at a comprehensive ADHD clinic at a tertiary hospital in urban Oman. Sociodemographic characteristics, clinical presentations, and factors associated with employment status were collected. 41% of the participants were actively seeking employment. Among the correlates of unemployment is obesity (OR 5.64, *p* = 0.011, 95% CI 1.49–21.43). Other variables, including education level and marital status, also influenced employment rates, with bachelor’s degree holders showing higher chances of unemployment (OR 5.35, *p* = 0.009, 95% CI 1.52–18.88). Marital status was closely associated with unemployment, with 39.5% of married individuals unemployed (*p* = 0.022). Furthermore, anxiety disorders were closely associated with unemployment (*p* = 0.026). Nearly one-third of the cohort had a comorbid substance use disorder (30%), and 6% reported suicidal attempts. This study highlights the significantly high prevalence of job seeking among adults with ADHD. Obesity, education level, marital status, and anxiety disorders were strongly associated with job search.

## 1. Introduction

Attention deficit hyperactivity disorder (ADHD) has been widely documented to be on the rise in many parts of the world ([1]). As detailed in most international psychiatric nomenclatures such as the International Classification of Diseases (ICD) and the Diagnostic and Statistical Manual of Mental Disorders (DSM), ADHD manifests itself through persistent inattention and hyperactivity, sometimes accompanied by impulsivity, leading to notable functional limitations and a reduced quality of life ([5]).

Although commonly identified during childhood, it affects approximately 3 to 5% of children and adolescents ([29]). Recent studies suggest that a significant portion of people with ADHD continue to experience its primary symptoms well into adulthood, with global prevalence rates estimated to be around 2.5% among adults ([31]). However, data on the clinical presentation and treatment outcomes of adults with ADHD in Oman remain limited. A recent prospective study conducted among adults with ADHD demonstrated high improvement rates, particularly with methylphenidate, and identified predictors of treatment response ([4]).

Although sociodemographic factors have been widely explored in adult ADHD, little has been reported about employment in countries of the Arabian Gulf, and Oman is no exception. The structure of the Omani population is pyramidal, with a significant number of young adults, likely leading to high competition to secure jobs. It remains to be seen how adults with ADHD fare in this endeavor. The unemployment rate among people between the ages of 15 and 24 is higher than previously speculated, at approximately 21% in Oman ([15]). The government initiative to ensure better job opportunities for job seekers has been ambitious, with the aim of creating more job opportunities. However, it is not clear whether these jobs are accessible to adults with ADHD. To decontextualize job seekers with ADHD, studies are needed to chart factors associated with job seeking among adults with ADHD. This has the potential to identify the barriers they face in their search for employment and lead to evidence-based prevention and mitigation strategies. Specifically, rather than exclusively focusing on symptom reduction, which is common practice, a concerted effort may be needed to introduce functional life skills to adults with ADHD. This could improve their employability and overall quality of life ([16]).

There is an indication in the literature to suggest that there is an increased risk of unemployment and marginalization in the labor market among adults with ADHD compared to those without the condition ([14]). Helgesson et al. ([14]) examined the factors that contribute to finding and keeping a job for what they called ‘labor market marginalization’ (LMM) among people with ADHD. This study found that 61% of the Sweden cohort were markedly low in LMM after a diagnosis of ADHD. According to the ‘*Consensus Statement on How ADHD Affects Employment*’ reported by [2] ([2]), there are no adequate provisions to protect against occupational difficulties among people with ADHD. In contrast, Li et al. ([19]) recruited 25,358 individuals with ADHD, who were born between 1958 and 1978 and were 30 to 55 years old during the study period (1 January 2008 to 31 December 2013) in Sweden. This study suggests that individuals taking ADHD pharmacotherapy have a lower risk of subsequent long-term unemployment.

With an emerging view that there is a higher prevalence of job search among adults with ADHD compared to the general population ([13]), the natural extension of this is that people with ADHD tend to exacerbate the challenges faced by people with ADHD and their families ([13]). Although the precise mechanisms that link ADHD and job search remain incompletely understood, several factors have been speculated that contribute to the predicament of adult ADHD and employment. First, ADHD may be exacerbated by the stigma attached to the condition, leading to discrimination and limited opportunities at work ([24]; [26]). Intimately related to the symptoms of ADHD are dispositions such as attention loss ([6]), memory loss ([6]), poor self-regulation ([30]), and time management problems ([20]), which are likely to be perceived as ‘bad manner’, a feat that is likely to lead to disapproval from management. Second, a study in Oman found that children with ADHD are apt to drop out of school ([22]), which has a net adverse effect on educational attainment, and, as an adult, this can set the precedent for poor employment skills.

The objective is to study to decipher the contributing factors of unemployed people, which, in turn, is likely to lay the groundwork for the establishment of remedial services for people with ADHD. Understanding the underlying factors of this association could facilitate the development of more effective interventions aimed at reducing the socioeconomic impact of the disorder and improving the lives of affected individuals and their families. There is a limited understanding of the correlation between ADHD among Omanis and their entry into the labor force. Prolonged job search is associated with economic hardships, decreased mental and physical health, and increased mortality rates ([3]). The primary objective of the study is to investigate the relationship between ADHD in adults and unemployment in Oman. Until a further strategy to conduct a more robust prospective and longitudinal study is in place, this sentinel study aims to capitalize on medical records in order to deduce whether there is a link between the presence of ADHD and unemployment. Using a retrospective design, our objective was to discern the prevalence of unemployment among people diagnosed with ADHD and to identify potential contributing factors.

Study Hypotheses

Patients with higher BMI (pre-obesity or obesity) would have higher odds of unemployment compared with patients in the normal BMI range.Lower educational level would be associated with higher odds of unemployment.Comorbid psychiatric conditions (depression, anxiety, personality disorders, substance use disorder) would be associated with higher unemployment rates.Sociodemographic factors (age, sex, marital status) would show significant associations with employment status.

## 2. Materials and Methods

### 2.1. Study Design and Setting

This retrospective cohort study was conducted at the Department of Behavioral Medicine at SQUH in Muscat, Oman. The study included adult patients diagnosed with ADHD in an outpatient setting. The Oman healthcare system is classified into primary, secondary, and tertiary healthcare. Tertiary centers tend to offer complementary clinical services that in some cases may not be available in the primary or secondary setting. Therefore, the present cohort constituted the only comprehensive national service for adults with ADHD located in a teaching hospital affiliated with Sultan Qaboos University.

### 2.2. Inclusion and Exclusion Criteria of Study Participants

The study included adult patients diagnosed with ADHD who met the ADHD criteria according to ICD-11 ([36]) and the DSM-5 criteria ([5]). Participants with an intellectual disability were excluded to ensure that the findings specifically reflect the impact of ADHD on job-seeking without the confounding effects of intellectual impairments.

### 2.3. Case Definition and Ascertainment

This study included men and women aged 18 to 60 years who attended the Adult ADHD Clinic from its inception in January 2018 to December 2022.

Routine procedures for patients with ADHD seeking consultation from the current adult ADHD clinic are as follows. The initial interview involved symptom review, and history-taking. This is followed by evaluations using rating scales or checklists (e.g., Conners’ Adult ADHD Rating Scales or Adult ADHD Self-Report Scale-V1.1).

Given the high comorbidity in adults with ADHD, evaluations of adult ADHD are often performed separately from acute episodes to ensure precision. This also involved a medical examination. Suppose the respondent appears to have met the initial screening for adult ADHD. In that case, he/she participates in the semi-structured interview using the style and format of the Composite International Diagnostic Interview (CIDI) for which a senior clinical member is trained to perform. The CIDI has been developed to provide practitioners with a reliable and valid assessment of specific and general psychological disorders according to the definitions and criteria of ICD-11 (17) and DSM-5 ([5]). 

In addition to the diagnosis, the following sociodemographic and clinical factors were extracted from medical records, including age, sex, marital status, and the highest educational level achieved. Regarding the risk, BMI was also sought. In the hospital system, BMI is calculated according to the protocol defined by the World Health Organization (WHO) ([33]). Accordingly, the BMI is calculated by dividing an individual’s weight in kilograms by the square of their height in meters. Using an established WHO classification, the following BMIs were adhered to in the present study: Underweight (BMI less than 18.5); Normal weight (BMI between 18.5 and 24.9); Pre-obesity (BMI between 25.0 and 29.9); Obesity class I (BMI between 30.0 and 34.9). Mathematically, the formula is expressed as BMI = weight (kg)/height (m^2^).

In addition to risk factors such as weight variation, the topology of ADHD was defined according to DSM-5 ([5]): as predominantly inattentive presentation, predominantly hyperactive-impulsive presentation, or combined presentation. Other variables include a family history of ADHD and a history of substance use disorder. Substance Use Disorder, as defined by the DSM-5, includes various types of substance-related issues. Examples include Alcohol Use Disorder, characterized by excessive drinking and impaired control; Cannabis Use Disorder, involving problematic marijuana use leading to negative consequences; Opioid Use Disorder, which pertains to the misuse of opioids like prescription pain relievers or heroin; and Tobacco Use Disorder, marked by dependence on nicotine products. Each of these disorders can significantly impact an individual’s functioning and well-being. The co-morbid psychiatric disorders sought included bipolar affective disorder, major depressive disorder, anxiety disorders, and personality disorders. Other variables sought from the medical record are the history of scholastic skill problems, according to the ICD-10 classification of mental and behavior disorders ([35]), encompassing specific spelling disorders, specific arithmetical skills disorders, mixed scholastic skills disorders, other scholastic skills disorders, developmental disorders of scholastic skills, and unspecified conditions, all grouped under the term “History of Specific Developmental Disorders of Scholastic Skills.” Other variables included are Suicidal Attempts and Forensic History. The latter involved an investigation of whether the participant had a history of misconduct with the law.

### 2.4. Data Source and Collection

Data were collected from hospital information system medical records and entered into an excel data collection sheet. All extracted variables were entered into an Excel sheet using predefined coding. Categorical variables (e.g., sex, ADHD presentation type, presence of substance use disorder, comorbid psychiatric diagnosis, forensic history, suicidal attempt history, and scholastic skill disorder) were coded as binary (0 = absent, 1 = present) or as categorical numeric codes where applicable (e.g., ADHD subtype: 1 = inattentive, 2 = hyperactive–impulsive, 3 = combined). Continuous variables such as age and BMI were entered as recorded and later categorized based on WHO classifications.

### 2.5. Statistical Analysis

The data collected were managed and statistical analyses were performed using IBM SPSS Statistics 29.0. Continuous variables were described using mean with standard deviation, and categorical variables were presented using frequency with percentages. Bivariate categorical comparisons between employment status, demographics, and clinical characteristics were made using the chi-square test. Continuous information was compared between groups using an independent *t*-test. The analyzer performed an adjusted logistic regression model with unemployment as a dependent variable, and the predictors included in the model were variables that showed a *p* value < 0.20 in the bivariate analysis. A *p*-value < 0.05 was considered statistically significant.

### 2.6. Ethical Approval

Ethical approval for the study was granted by the Medical Research Ethics Committee (MREC) of the College of Medicine and Health Sciences, Sultan Qaboos University, Muscat, Oman (MREC #2260) on 30th of April 2023. The study adhered to the principles outlined in the Declaration of Helsinki for ethical research involving human subjects ([37]).

## 3. Results

In this study, a cohort of 179 patients diagnosed with adult ADHD was examined to determine the factors associated with job search. The sociodemographic characteristics of the study population are presented in Table 1. The mean age of the cohort was 25.79 ± 6.76 years, and the mean BMI was 25.20 ± 6.14, with almost half of the patients within the average BMI range. Most of the cohort was male, 61%, and 18% were married. Inattention was the most common clinical presentation, accounting for 65% of the cases, while a quarter of the patients presented mixed types of hyperactivity and inattention, and a family history of ADHD was reported in 23% of the cases. Substance use disorder (SUD) was the most associated comorbidity among patients with ADHD present in 30% of cases. Other comorbidities, such as anxiety were found in 26%, depression in 18%, bipolar affective disorder, and personality disorders affected 15% of patients, and a history of specific developmental disorders of scholastic skills was found in 7% of the cases. Almost three-quarters of the cohort exhibited at least one comorbidity. Suicidal attempts were reported in 6% of the patients.

Students were excluded from the analysis of unemployment. A total of 105 patients were included in the univariate and multivariate comparisons and 41% (n = 43) were unemployed. Univariate analysis as shown in Table 1 revealed that obese patients had a higher risk of unemployment, with 40% of unemployed patients being obese, compared to only 13% of employed patients (*p* = 0.022). The level of education also showed a borderline association with job-seeking (*p* = 0.053), In addition, marital status was significantly associated with unemployment, with 39.5% of married patients being unemployed (*p* = 0.012). None of the other variables were statistically associated with job search. The multivariate logistic regression model as shown in Table 2 was significant (*p* < 0.001), with Cox & Snell R2 explaining 27% of the variations. Obesity was identified as an independent predictor of unemployment (OR 5.64, *p* = 0.011, 95% CI 1.49–21.43), along with people holding a bachelor’s degree (OR 5.35, *p* = 0.009, 95% CI 1.52–18.88).

## 4. Discussion

One of the zeitgeists that increasingly encroaches on the persisting ‘medical paradigm is whether those who are marked with ADHD should be labeled as having neurodevelopmental disorder or simply be considered as neurodivergent ([9]). The core view of those who advocate ADHD as a neurodivergence posit that there is an undeniable fact that there are variations in cognitive, psychological, and behavioral traits. Thus, rather than focus on their pathology, it should be prudent to focus, as in the case of people with ADH, on their unique strengths and abilities and, at the same time, seek the necessary accommodations, support, and provide remedial services to improve their functionality and the result meaningful existence. The emerging narrative that inclusive education, the sociological concept of medicalization, and indeed teaching are intimately linked to nidotherapy are likely to be part and parcel of the emerging movement on neurodiversity ([27]; [28]). If the emerging zeitgeist has heuristic value, then it should be expected that adults with ADHD will be accommodated in the labor force. In fact, according to the Disability Act, people with ADHD constitute ‘neurodiverse’ and therefore are liable for protection reserved for people of special needs and talents ([18]). Therefore, any misinterpretation directed at ADHD would constitute workplace discrimination ([18]). However, such provisions have been limited to a few countries in the world. But with an emerging interest in recasting our view towards those who are different from neurotypical, there is a chance such a paradigm will also pervade countries in the global south. With this background, this study has begun to examine the factors associated with unemployment in Oman.

Being a sentinel study, this study focuses on attendees attending a dedicated adult ADHD clinic. The sociodemographic profile of the study population reflects the various characteristics commonly observed among people diagnosed with adult ADHD. The average age of 25.79 years is consistent with previous research that indicates, though not directly explored in this study, that ADHD symptoms often persist into adulthood and there is a suggestion that disposition tends to affect individuals throughout their life ([8]; [21]). Consistent with other clinical topologies of ADHD, the present cohort consisted predominantly of male participants, that is, men comprise 61% of the cohort ([23]). Regarding the variation in symptoms of ADHD, inattention emerged as the most prevalent symptomatology among the present attendees, accounting for 65% of cases. The literature generally suggests that inattention is more commonly observed in adults with ADHD than in the other subtypes ([34]). In contrast, hyperactivity is often more prominent in childhood ADHD presentations, but its severity has been indicated to be depleted with increased maturity and therefore hyperactivity decreases significantly at the onset of adulthood ([34]). The incremental decrease in hyperactivity with added maturity echo what has been documented in neurobiological studies: the frontal lobe, particularly the prefrontal cortex, plays a crucial role in executive functions such as planning, decision-making, impulse control, and working memory. As the critical brain area involved in executive function matures, typically during adolescence and early adulthood, studies have suggested that there is a notable attenuation in executive function symptoms associated with conditions such as ADHD ([32]).

A substantial proportion of people presented with comorbidities, which can be attributed to shared neurobiological mechanisms, including changes in neurotransmitter systems and genetic susceptibility ([17]). Developmental disturbances, environmental influences, and the impaired executive functioning characteristic of ADHD contribute to the increased risk of developing comorbid psychiatric disorders ([17]). In addition, challenges in diagnosis and treatment can arise due to overlapping symptoms and complexities of treatment ([17]). This emphasizes the importance of comprehensive evaluations and treatment approaches to address the diverse needs of individuals with ADHD and comorbidities ([17]). SUD was notably prevalent in our cohort, present in 30% of cases, underscoring the increased risk of substance abuse among people with ADHD. This finding is consistent with previous studies indicating that SUD affects approximately one in five adults with ADHD ([17]). Such a high prevalence warrants targeted interventions and integrated treatment approaches. The increase in the occurrence of SUD among adults with ADHD can be attributed to several interconnected factors, including the hypothesis of self-medication, in which people with ADHD can use substances to alleviate symptoms ([10]).

The 30% of patients with ADHD in our cohort who reported substance abuse issues do not have legal problems, as evidenced by only 1.7% having a forensic history. This suggests that substance use can manifest in ways that do not lead to legal consequences. Many individuals may use substances to self-medicate for ADHD symptoms or co-occurring mental health issues ([10]) without engaging in illegal activities. Factors such as personal circumstances, social support, and access to treatment also influence this relationship.

Furthermore, suicidal attempts were documented in 6% of the patients among the study participants. These occurrences could potentially be attributed to neurobiological vulnerabilities, including impulsivity and risk-taking behaviors, often associated with adult ADHD. Furthermore, research indicates that financial distress is linked to a four-fold higher risk of suicide among people with ADHD ([7]). Considering that adults with ADHD often face unemployment and difficulty maintaining employment ([7]), these factors may contribute to the increased prevalence of such attempts.

Analysis of employment status within our cohort revealed that 41% (n = 43) were unemployed. Univariate analysis showed that pre-obese/obese patients had a higher risk of job search, 40% of unemployed patients were pre-obese/obese, compared to only 13% of employed patients (*p* = 0.002). The level of education was closely associated with unemployment status (*p* = 0.053), with 39.5% of married patients unemployed (*p* = 0.022). Married individuals may be more represented in the unemployed sample due to sociocultural factors. For example, family responsibilities often limit their job-seeking opportunities, as they may prioritize caregiving and household duties over employment ([25]). None of the other variables were statistically associated with job-seeking. The multivariate logistic regression model was significant (*p* < 0.001), with Cox & Snell R2 explaining 27% of the variations.

The increased risk of job searching among pre-obese and obese individuals may be due to various interrelated factors. Obesity can cause physical health problems, such as reduced mobility, chronic pain, and fatigue, which can limit the ability of people to perform work tasks effectively ([12]). Furthermore, weight bias and discrimination on the job can contribute to hiring disparities and hinder career advancement opportunities for obese individuals ([11]). Psychosocial factors, including low self-esteem, depression, and social stigma, can also affect job performance and interpersonal relationships in work settings ([11]). Furthermore, obese individuals may face challenges in accessing educational and vocational training programs, exacerbating socioeconomic disparities, and perpetuating the cycle of unemployment ([11]). In Omani culture, views on obesity can be complex and multifaceted. While there may be a tendency to associate higher body weight with prosperity and health in some contexts, obesity is increasingly recognized as a public health concern, with rising awareness of its associated health risks. Consequently, there is a growing negative perception towards obesity, particularly as it relates to lifestyle diseases. Addressing these multifaceted barriers through targeted interventions and workplace accommodations is essential for promoting inclusive employment opportunities and improving outcomes for pre-obese/obese individuals with adult ADHD in the workforce.

Psychiatrists can play a critical role in the care of people with adult ADHD and obesity, offering assessment, diagnosis, and customized treatment plans that may include pharmacological interventions, psychotherapy, and lifestyle modifications. They address co-morbid conditions, offer patient and family education, and provide emotional support. A systematic review and meta-analysis found a significant association between ADHD and disordered eating or addictive-like eating behaviors, with several studies suggesting that negative affectivity or emotion dysregulation may mediate these behaviors ([12]). Additionally, psychiatrists advocate for policy changes and public health initiatives to improve access to care and promote healthier environments. With their experience and comprehensive approach, psychiatrists aim to improve the well-being and quality of life of people who manage adult ADHD and obesity.

Similarly, our study revealed that people with a bachelor’s degree were more likely to be unemployed, contradicting conventional expectations regarding the positive correlation between higher education and employment status ([39]). This unexpected finding underscores the need for more research on the relationship between educational attainment, ADHD symptomatology, and employment outcomes. The higher likelihood of job searching among individuals with bachelor’s degrees in our study could stem from various factors. One possibility is the saturation of certain job markets, where the supply of qualified candidates exceeds demand, leading to increased competition for employment opportunities ([38]). Moreover, employers may prefer high school graduates due to the associated lower costs of employment, as well as the expectation that these individuals possess essential skills for the workforce. Thus, educational background plays a significant role in employment opportunities within the Omani context.

### Strengths and Limitations

While our study provides valuable insights into job-seeking behaviors among adults with ADHD, several limitations should be acknowledged. The retrospective design limits the ability to draw causal inferences, and the sample size may affect generalizability. However, the study has several strengths. First, it is one of the pioneering efforts to examine the association between ADHD and job-seeking behavior in Oman, addressing an important research gap. Additionally, the study involved comprehensive data collection, including detailed patient information on clinical history, demographics, and job-seeking behavior, which allowed for a thorough analysis of factors associated with these outcomes. Furthermore, the sample includes all patients from the only adult ADHD clinic in Oman, enhancing generalizability by reflecting a clinical population representative of adults seeking care for ADHD in the region. Future longitudinal studies could further explore the relationship between ADHD, job-seeking, and related factors, offering a more comprehensive understanding of these variables.

## 5. Conclusions

In conclusion, this study sheds light on the pressing issue of weight-related concerns and job seekers among adults with ADHD in the Arabian Gulf country, Oman. A substantial proportion (40%) of the participants identified as pre-obese or obese. Furthermore, our findings highlight the multifaceted nature of job search among adults with ADHD, pre-obesity/obesity was identified as an independent predictor of job search. Additionally, educational attainment played a critical role, as individuals holding a bachelor’s degree exhibited a higher likelihood of unemployment. Marital status also influenced employment rates, with a notable percentage of married individuals being unemployed. Furthermore, anxiety disorders were significantly associated with unemployment. These findings underscore the necessity for targeted interventions to address the unique challenges faced by adults with ADHD, particularly concerning obesity, educational support, and mental health, to enhance their employability and overall quality of life.

## Figures and Tables

**Table 1 behavsci-16-00067-t001:** Distribution of demographic, clinical characteristics, and association with job-seeking.

Variables	Total	Job-Seeking		*p*-Value
		No (n = 62)	Yes (n = 43)	
Age	25.79 ± 6.76	27.60 ± 7.28	29.65 ± 7.20	0.156
Gender				
Female	70 (39.1)	24 (38.7)	22 (51.2)	0.234
Male	109 (60.9)	38 (61.3)	21 (48.8)	
BMI				
Underweight	18 (11.3)	7 (12.7)	4 (10.0)	0.022
Normal	74 (46.3)	29 (52.7)	13 (32.5)	
Pre-obesity	32 (20.0)	12 (21.8)	7 (17.5)	
Obesity class I and above	36 (22.5)	7 (12.7)	16 (40.0)	
Marital status				
Single	146 (81.6)	52 (83.9)	26 (60.5)	0.012
Married	33 (18.4)	10 (16.1)	17 (39.5)	
The highest educational level attained				
Less than high school	16 (9.2)	3 (5.1)	3 (7.0)	0.053
High school	72 (41.6)	9 (15.3)	8 (18.6)	
Bachelor’s degree	41 (23.7)	15 (25.4)	20 (46.5)	
Master’s and Professionals	44 (25.4)	32 (54.2)	12 (27.9)	
Clinical presentation				
Hyperactivity	19 (10.7)	10 (16.1)	8 (18.6)	0.925
Inattention	115 (64.6)	44 (71.0)	29 (67.4)	
Combined type	44 (24.7)	8 (12.9)	6 (14.0)	
Family history of ADHD				
Yes	41 (23.0)	11 (17.7)	5 (11.6)	0.425
No	137 (77.0)	51 (82.3)	38 (88.4)	
Substance use disorder				
Yes	54 (30.3)	11 (17.7)	8 (18.6)	1.000
No	124 (69.7)	51 (82.3)	35 (81.4)	
Bipolar affective disorder				
Yes	27 (15.2)	10 (16.1)	11 (25.6)	0.321
No	151 (84.8)	52 (83.9)	32 (74.4)	
Personality disorders				
Yes	28 (15.7)	11 (17.7)	10 (23.3)	0.621
No	150 (84.3)	51 (82.3)	33 (76.7)	
Depressive disorders				
Yes	32 (18.0)	10 (16.1)	5 (11.6)	0.582
No	146 (82.0)	52 (83.9)	38 (88.4)	
Anxiety Disorders				
Yes	47 (26.4)	21 (33.9)	7 (16.3)	0.071
No	131 (73.6)	41 (66.1)	36 (83.7)	
Any psychiatric comorbidities				
Yes	131 (73.6)	40 (64.5)	27 (62.8)	1.000
No	47 (26.4)	22 (35.5)	16 (37.2)	
History of Specific Developmental Disorders of Scholastic Skills				
Yes	13 (7.3)	4 (6.5)	2 (4.7)	1.000
No	165 (92.7)	58 (93.5)	41 (95.3)	
Suicidal attempt				
Yes	10 (5.6)	2 (3.2)	2 (4.7)	1.000
No	168 (94.4)	60 (96.8)	41 (95.3)	
Forensic History				
Yes	3 (1.7)	1 (1.6)	-	1.000
No	175 (98.3)	61 (98.4)	43 (100.0)	

**Table 2 behavsci-16-00067-t002:** Multivariate logistic regression analysis employed vs. unemployed.

Variables	Beta	S.E.	Sig.	Odds Ratio	95% CI of Odds Ratio
Age	−0.03	0.05	0.573	0.971	0.88–1.08
Marital status—Married	1.62	0.87	0.063	5.048	0.92–27.78
The highest educational level attained					
<High school	1.18	0.95	0.214	3.257	0.51–20.94
High school diploma	1.01	0.78	0.193	2.757	0.60–12.68
Bachelor’s degree	1.68	0.64	0.009	5.365	1.52–18.88
BMI					
Pre-obesity	0.21	0.67	0.753	1.235	0.33–4.59
Obesity class I and above	1.73	0.68	0.011	5.641	1.49–21.43
Anxiety Disorders	−1.55	0.70	0.026	0.212	0.05–0.83

## Data Availability

The data presented in this study are available on request from the corresponding author. The data are not publicly available to protect patient privacy.

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
