# Peer review of "The Association Between Job-Seeking & Adult Attention Deficient Hyperactivity Disorder Among Working-Age Individuals in Oman: A Retrospective Study from the National Clinical Service for Adult ADHD in Oman"

_behavsci, 2026, doi:10.3390/bs16010067_

Round 1
Reviewer 1 Report
Comments and Suggestions for Authors
The study addresses a relevant topic and provides data in a context in which information about neurodivergence and work situation is not extensively published.
The rationale of the study is clear and the methods are well described. Maybe there is too much information on some technical details that are not typically included in scientific paper. For example section 2.5 on Statistical analyses provides too much information, it is enough to mention the software and the key procedures used.
My main concern with the study is the fact that inferential statistics are used and no specific hypotheses are formulated. This is a fundamental flaw that needs correcting. Regression analyses and t tests, ANOVA procedures are inferential and they require specific hypotheses.
This shortcoming leads me to the next remark. The directionality of hypotheses is not clear. The paper for example states: "Obesity was identified as an independent predictor of unemployment (OR 5.64, p=0.011, 95% CI 1.49 – 21.43), along with people holding a
bachelor's degree (OR 5.35, p=0.009, 95% CI 1.52 – 18.88)." Is it the case that obesity leads to unemployment as stated or unemployment increases the risk of obesity? Based on cross-sectional data the directionality of this claim cannot be supported. This is just one of the examples that can be found in the interpretation of the results. One could find reasonable arguments for the association between BA and employment - although the effect is positive, which is weird ... I would expect that higher education decreases the risk of unemployment.
The following section does not make sense as it conflates education with marital status: " The level of education was closely associated with job-seeking (p=0.053), with 39.5% of married patients were unemployed (p=0.012). "
I also think the sample is rather small to draw definite conclusions concerning the ADHD and job seeking. Research previously published in Behavioral Sciences (doi.org/10.3390/bs15040420) shows that self-diagnosis could also be used to select participants and increase the sample size in order to get a more accurate understanding of job seeking in people with ADHD.
Data collection was not clear. Which data was collected and how were the questions phrased? The paper states: Data were collected from hospital information system medical records and entered into an excel data collection sheet. No additional information is provided on the way variables were coded. Were all variables based on diagnostic elements? It is not clear how data was generated for the analyses.
The use of AI In writing should be acknowledged. Phrases such as: "One of the zeitgeists that increasingly encroaching on the persisting ‘medical paradigm is whether those who are marked with ADHD should be labeled as having neurodevelopmental disorder or simply be considered as neurodivergent ", are weird and contorted.
Author Response
The study addresses a relevant topic and provides data in a context in which information about neurodivergence and work situation is not extensively published.
Author Response:
Thank you for your valuable comments and suggestions.
The rationale of the study is clear and the methods are well described. Maybe there is too much information on some technical details that are not typically included in scientific paper. For example section 2.5 on Statistical analyses provides too much information, it is enough to mention the software and the key procedures used.
Author Response:
Thank you for this valuable comment. We have edited the Statistical Analysis section by removing unnecessary technical detail and retaining only the key procedures and software used.
My main concern with the study is the fact that inferential statistics are used and no specific hypotheses are formulated. This is a fundamental flaw that needs correcting. Regression analyses and t tests, ANOVA procedures are inferential and they require specific hypotheses.
Author Response:
Thank you for this insightful comment. The revised text is now included in the manuscript and highlighted in yellow.
This shortcoming leads me to the next remark. The directionality of hypotheses is not clear. The paper for example states: "Obesity was identified as an independent predictor of unemployment (OR 5.64, p=0.011, 95% CI 1.49 – 21.43), along with people holding a
bachelor's degree (OR 5.35, p=0.009, 95% CI 1.52 – 18.88)." Is it the case that obesity leads to unemployment as stated or unemployment increases the risk of obesity? Based on cross-sectional data the directionality of this claim cannot be supported. This is just one of the examples that can be found in the interpretation of the results. One could find reasonable arguments for the association between BA and employment - although the effect is positive, which is weird ... I would expect that higher education decreases the risk of unemployment.
Author Response:
Thank you for this important observation. We fully agree that, given the cross-sectional and retrospective nature of our study, causality or directionality cannot be inferred. Our intention was to describe statistical associations rather than suggest that obesity or educational level lead to unemployment. We have revised the Results and Discussion sections to clarify that these findings reflect associations only, without implying causal pathways.
The following section does not make sense as it conflates education with marital status: " The level of education was closely associated with job-seeking (p=0.053), with 39.5% of married patients were unemployed (p=0.012). "
Author Response:
Thank you for pointing this out. We agree that the sentence was unclear and incorrectly combined two separate variables (education and marital status). We have now revised the Results section.
I also think the sample is rather small to draw definite conclusions concerning the ADHD and job seeking. Research previously published in Behavioral Sciences (doi.org/10.3390/bs15040420) shows that self-diagnosis could also be used to select participants and increase the sample size in order to get a more accurate understanding of job seeking in people with ADHD.
Author Response:
Thank you for this insightful comment. We agree that the sample size limits the generalizability of our findings. We have added this point to the limitations section & highlighted in yellow in the manuscript.
Data collection was not clear. Which data was collected and how were the questions phrased? The paper states: Data were collected from hospital information system medical records and entered into an excel data collection sheet. No additional information is provided on the way variables were coded. Were all variables based on diagnostic elements? It is not clear how data was generated for the analyses.
Author Response:
Thank you for this important comment. We have expanded and clarified this section accordingly, and the revisions are highlighted in yellow in the manuscript.
The use of AI In writing should be acknowledged. Phrases such as: "One of the zeitgeists that increasingly encroaching on the persisting ‘medical paradigm is whether those who are marked with ADHD should be labeled as having neurodevelopmental disorder or simply be considered as neurodivergent ", are weird and contorted.
Author Response:
Thank you for this valuable observation. We would like to clarify that the paragraph in question was written by one of the co-authors, a senior professor of psychology with extensive expertise in this field. The phrasing reflects his academic voice and stylistic choices.
Reviewer 2 Report
Comments and Suggestions for Authors
Dear authors:
After reviewing the article submitted to the journal Behavioral Sciences, I must say that you have presented a very interesting piece of work. This article contributes important findings for both the scientific community and society in general. However, I would like to make a few suggestions:
- The title is long and I believe it could be shortened so that it is not repetitive. ‘Retrospective study on the association between job search and adults with ADHD’ or ‘Job search in adults with ADHD: a retrospective study’. Please consider this if you deem it appropriate.
- The abstract is coherent and well structured. All the information provided is important and allows for an understanding of the main results of the article. However, it exceeds the 200-word limit. It should be shortened.
- I do not consider the keywords ‘obesity’ and ‘overweight’ to be relevant, as they are only variables analysed. In this sense, marital status, educational level or anxiety disorders would also have to be included, but this would not make sense. I believe that the keywords that should remain are ADHD, Adult, Employment status and Job search.
- The introduction should cite the 2022 DSM-5-TR, as it is the most recent manual.
- The introduction could benefit from including more recent studies (2021-2025). I also think it is too short. I suggest expanding it by adding more previous studies.
- The objective of the research is clear and consistent.
- The Method section is very well defined. The steps for data collection and subsequent analysis are clearly written.
- Please note that when a number is followed by a percentage sign, there should be a space between the number and the sign (5%). Please check this throughout the article.
- Remove the yellow shading from the tables. To indicate significance, you can highlight the number in bold, but not in yellow.
- The results section is well structured, with no errors of interpretation. The tables are necessary and add value to the research work carried out.
- The discussion, presentation of limitations and strengths, and conclusions are presented in an orderly manner. The results are discussed firmly and supported by recent previous studies.
- The references section does not comply with MDPI guidelines. Please review the journal's guidelines in detail and correct this section.
Overall, this is a valuable piece of work, but it could be improved by revising the title, abstract, keywords, introduction, and some of the sections mentioned above.
Author Response
After reviewing the article submitted to the journal Behavioral Sciences, I must say that you have presented a very interesting piece of work. This article contributes important findings for both the scientific community and society in general. However, I would like to make a few suggestions:
Author Response:
Thank you for your positive feedback and thoughtful comments. We appreciate your recognition of the contribution of our work. We have carefully considered your suggestions and revised the manuscript accordingly.
- The title is long and I believe it could be shortened so that it is not repetitive. ‘Retrospective study on the association between job search and adults with ADHD’ or ‘Job search in adults with ADHD: a retrospective study’. Please consider this if you deem it appropriate.
Author Response:
Thank you for this helpful suggestion. We agree that a shorter title would improve clarity; however, the journal’s guidelines advise against using abbreviations in the title. For this reason, we have opted to retain the original title.
- The abstract is coherent and well structured. All the information provided is important and allows for an understanding of the main results of the article. However, it exceeds the 200-word limit. It should be shortened.
Author Response:
Thank you for noting this. We have revised the abstract to ensure it meets the 200-word limit.
- I do not consider the keywords ‘obesity’ and ‘overweight’ to be relevant, as they are only variables analysed. In this sense, marital status, educational level or anxiety disorders would also have to be included, but this would not make sense. I believe that the keywords that should remain are ADHD, Adult, Employment status and Job search.
Author Response:
Thank you for this hlepful comment . We have revised the keywords accordingly and retained only the relevant terms.
- The introduction should cite the 2022 DSM-5-TR, as it is the most recent manual.
Author Response:
Thank you for this suggestion. We acknowledge the publication of the DSM-5-TR in 2022. However, the ethical approval and clinical diagnostic procedures for this study were based on the DSM-5, which was the version in use during the study period. Since the diagnostic criteria for ADHD remain unchanged between DSM-5 and DSM-5-TR, we have opted to retain the original DSM-5 citation to accurately reflect the diagnostic framework used in the clinical records.
- The introduction could benefit from including more recent studies (2021-2025). I also think it is too short. I suggest expanding it by adding more previous studies.
Author Response:
Thank you for this helpful suggestion. We have revised and expanded the introduction to include a total of nine recent studies published between 2021 and 2025, thereby strengthening the background and contextual foundation of the manuscript.
- The objective of the research is clear and consistent.
Author Response:
Thank you for your positive feedback.
- The Method section is very well defined. The steps for data collection and subsequent analysis are clearly written.
Author Response:
Thank you for your positive feedback. We appreciate your recognition of the clarity and structure of the Methods section.
- Please note that when a number is followed by a percentage sign, there should be a space between the number and the sign (5%). Please check this throughout the article.
Author Response:
Thank you for the comment, we have adjusted this accordingly.
- Remove the yellow shading from the tables. To indicate significance, you can highlight the number in bold, but not in yellow.
Author Response:
Thank you for your feedback. We have removed the yellow highlight.
- The results section is well structured, with no errors of interpretation. The tables are necessary and add value to the research work carried out.
Author Response:
Thank you for your positive feedback.
- The discussion, presentation of limitations and strengths, and conclusions are presented in an orderly manner. The results are discussed firmly and supported by recent previous studies.
Author Response:
Thank you for your positive feedback. We appreciate your recognition of the clarity and strength of the discussion, limitations, and conclusions.
- The references section does not comply with MDPI guidelines. Please review the journal's guidelines in detail and correct this section.
Author Response:
Thank you for the comments. We have revised the references to follow Vancouver style in accordance with the MDPI guidelines.
Overall, this is a valuable piece of work, but it could be improved by revising the title, abstract, keywords, introduction, and some of the sections mentioned above.
Round 2
Reviewer 2 Report
Comments and Suggestions for Authors
Dear authors,
The changes made in the different sections have provided greater substance to the article. Some suggestions have not been taken into account but have been duly justified. Therefore, I consider that the article meets the desirable requirements of quality research.